# STPM_SAHI: A Small-Target Forest Fire Detection Model Based on Swin Transformer and Slicing Aided Hyper Inference

Ji Lin [1], Haifeng Lin [1,*] and Fang Wang [2,*]

1 College of Information Science and Technology, Nanjing Forestry University, Nanjing 210037, China
2 College of Electronic Engineering, Nanjing Xiaozhuang University, Nanjing 211171, China
* Correspondence: haifeng.lin@njfu.edu.cn (H.L.); wangfang0182217@njxzc.edu.cn (F.W.);
  Tel.: +86-25-8542-7827 (H.L.); +86-25-86175539 (F.W.)

**Abstract:** Forest fires seriously destroy the world's forest resources and endanger biodiversity. The traditional forest fire target detection models based on convolutional neural networks (CNNs) lack the ability to deal with the relationship between visual elements and objects. They also have low detection accuracy for small-target forest fires. Therefore, this paper proposes an improved small-target forest fire detection model, STPM_SAHI. We use the latest technology in the field of computer vision, the Swin Transformer backbone network, to extract the features of forest fires. Its self-attention mechanism can capture the global information of forest fires to obtain larger receptive fields and contextual information. We integrated the Swin Transformer backbone network into the Mask R-CNN detection framework, and PAFPN was used to replace the original FPN as the feature fusion network, which can reduce the propagation path of the main feature layer and eliminate the impact of down-sampling fusion. After the improved model was trained, the average precision ($AP_{0.5}$) of forest fire target detection at different scales reached 89.4. Then, Slicing Aided Hyper Inference technology was integrated into the improved forest fire detection model, which solved the problem that small-target forest fires pixels only account for a small proportion and lack sufficient details, which are difficult to be detected by the traditional target detection models. The detection accuracy of small-target forest fires was significantly improved. The average precision ($AP_{0.5}$) increased by 8.1. Through an ablation experiment, we have proved the effectiveness of each module of the improved forest fire detection model. Furthermore, the forest fire detection accuracy is significantly better than that of the mainstream models. Our model can also detect forest fire targets with very small pixels. Our model is very suitable for small-target forest fire detection. The detection accuracy of forest fire targets at different scales is also very high and meets the needs of real-time forest fire detection.

**Keywords:** forest fire detection; small-target forest fire; Swin Transformer; Slicing Aided Hyper Inference

## 1. Introduction

Forest, as a valuable resource for human survival, plays an important role in air purification, epidemic prevention, oxygen production, noise reduction, water conservation, and climate regulation [1]. However, more than 200,000 forest fires occur every year in the world, and a large area of forests are burned. Forest fires seriously destroy the world's forest resources and endanger biodiversity, and the detection of forest fires is very important, especially to detect early forest fires in time and prevent their spread. The traditional manual inspection has the disadvantages of having a long cycle, heavy workload, low efficiency, and dead angle of monitoring [2]. Sensors are also widely used in forest fire detection [3–5], including temperature sensors, smoke sensors, and infrared sensors. Although these sensors are sometimes effective, they have defects. First, a certain concentration of particles in the air must be reached to trigger the alarm. When the alarm is triggered, the fire may be too strong to be controlled, thus failing to achieve the purpose of early warning. Secondly,

most sensors can only work in a closed environment, not necessarily in large-scale forest areas. Third, when the concentration of non-fire particles reaches the alarm concentration of the sensor, the alarm sound will also be emitted automatically. In addition, remote sensing satellites are also used in the field of forest fire detection [6]. The monitoring range of satellites is wide and is not affected by environmental conditions such as terrain, but it is difficult to achieve continuous monitoring due to the limitation of repetition period and detection resolution, and poor mobility and flexibility. With the development of computer-vision technology, forest fire detection systems based on digital image-processing technology began to emerge. Image detection has the advantages of a short detection time, high accuracy, and flexible installation. It can be deployed on the watchtowers [7] or using UAVs [8,9] equipped with cameras to replace manual with automatic forest fire monitoring. Traditional fire detection methods based on computer-vision technology include feature extraction and color recognition of man-made forest fires. For instance, in [10], a color and disorder measurement method based on the RGB (red, green, blue) model was proposed to extract flame pixels from a video, which used color information to extract the flame pixels and dynamic information to verify the authenticity of the fire. Based on the study of numerous pictures and videos, Çelik [11] proposed a fuzzy color model, which extracted the flame region by using a statistical analysis method combined with motion analysis to improve the ability of the model to distinguish the flame target and the fire-like target. However, scholars have found that the artificially created flame features rely heavily on human prior knowledge, and the selection of features directly determines the performance of flame recognition. Moreover, the forest environment is complex and changeable, and the shape, texture, color, and scale of forest fires are also dynamic. Small-target forest fires are even more difficult to detect, which poses a great challenge to detecting them.

With the development of deep learning and the progress of computer hardware, convolutional neural network (CNN) has made many breakthroughs [12–14]. For example, one-stage target detection models include YOLO [15], SSD [16], EfficientDet [17], etc. The two-stage target detection models include RCNN [18], Fast-RCNN [19], and Mask-RCNN [20], based on Fast-RCNN, etc. Generally, two-stage object detectors are more accurate but slightly slower than one-stage detectors [21]. Impressed by the superior achievements of CNNs, many scholars have utilized them to perform forest fire detection. Zhang [22] created a forest fire identification model benchmark using Faster-RCNN, YOLO, and SSD to identify the flame. Kim [23] et al. used fast RCNN to extract flame and non-flame regions based on spatial features.

While CNNs can be used to extract local information about forest fires, they have limited ability in extracting remote features from global information, and their ability to deal with the relationship between visual elements and objects is weak. Inspired by Transformer's [24] use of self-attention in natural language processing (NLP) [25,26], many scholars proposed to use the self-attention mechanism to overcome the limitations of CNNs, and have achieved good results. The self-attention mechanism can obtain the relationship between remote elements more quickly, and pay attention to different areas of the image, thus integrating the information of the whole image. Vision Transformer (ViT) [27] is one of the state-of-the-art (SOTA) works in image recognition. Its self-attention mechanism enables it to outperform CNNs in image recognition. DETR [28] is the first method to use Transformer in target detection tasks. DETR adds a Transformer encoder and decoder to a standard CNN model, such as ResNet-50, and uses the set-matching loss function. Swin Transformer [29] is modified from ViT by using shifted windows (SW). It divides the fixed-size sample block in ViT into blocks (Windows) of different sizes according to their levels. Compared with ViT, Swin Transformer has a greatly reduced computational complexity and a linear computational complexity in the size of the input image. Swin Transformer has made breakthrough progress in the field of computer vision and has excellent performance in image classification and target detection tasks. Here, we introduce it into the field of forest fire detection and prove its effectiveness through experiments.

Another problem is that small-target forest fires are very small and lack sufficient details, so they are difficult to be detected. To solve this problem, we integrate our improved forest fire target detection model with Slicing Aided Hyper Inference (SAHI) [30] technology, which provides a pipeline of slice-aided reasoning for small-target forest fire detection and greatly improves the detection accuracy of small-target forest fires.

## 2. Materials and Methods

The contributions of this paper are as follows:

(1)　We use Swin Transformer to replace the original backbone (ResNet-50) of the Mask R-CNN to make full use of its self-attention mechanism, thereby obtaining a larger receptive field and context information by capturing global information and strengthening the ability of local information acquisition. The model pays more attention to and fully learns the characteristics of forest fires, which improves the detection performance of small-target forest fire images captured by cameras and has better performance in detecting intensive forest fires.

(2)　We use PAFPN [31] to replace the Mask R-CNN detection framework's original feature-fusion network, FPN [32]. Based on FPN, PAFPN adds a down-sampling module and an additional 3 × 3 convolution to build a bottom–up feature-fusion network to reduce the propagation path of the main feature layer and eliminate the impact of down-sampling fusion. The positioning capability of the whole feature hierarchy is enhanced.

(3)　After the model training, we integrated Slicing Aided Hyper Inference technology with our improved forest fire detection model, which solved the problem that the target pixels in small-target forest fires are small in proportion and lack sufficient details to be detected, while maintaining low computational resource requirements.

### 2.1. Dataset and Annotations

It is known to all that the performance of DL models depends largely on the quality and quantity of the training set. Therefore, in order to ensure that our model can learn the characteristics of various forest fires and have promising generalization ability, we used the forest fire data set created by the scientific research institute of Nanjing Forestry University. Meanwhile, we collected different types of aerial forest fire images from the Internet, including forest fires images taken by UAVs. The dataset is manually labeled fire areas and converted into COCO [33] format. We used 90% of the dataset for training and 10% for testing. Sample images in the training set are shown in Figure 1.

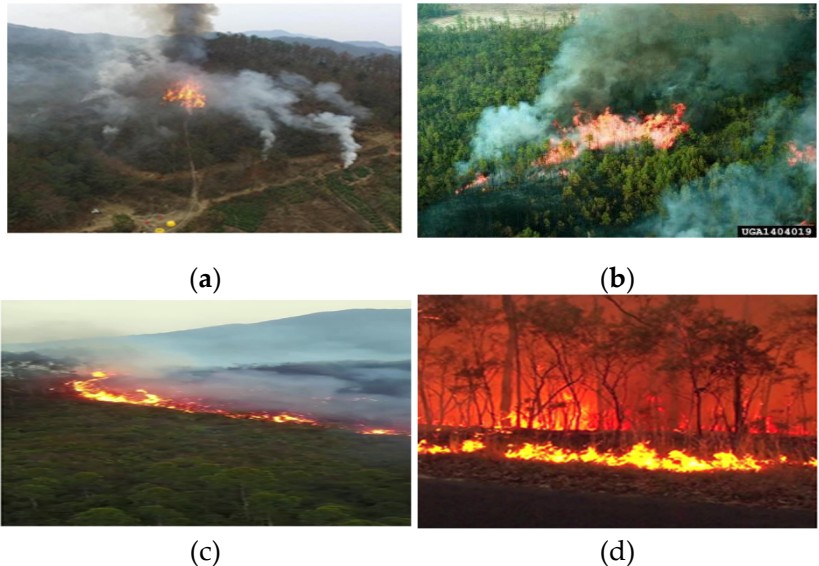

(**a**)　　　　　　　　　　　　　　　　　　　　(**b**)

(c)　　　　　　　　　　　　　　　　　　　　(d)

**Figure 1.** Sample images in the training set: (**a**–**d**) forest fire targets at different scales.

After learning the characteristics of the forest fires at different scales, the Slicing Aided Hyper Inference framework was integrated, and 332 small-target forest fire images were processed in the same way for testing to verify the detection accuracy of the improved model for small-target forest fires. Some pictures are shown in Figure 2.

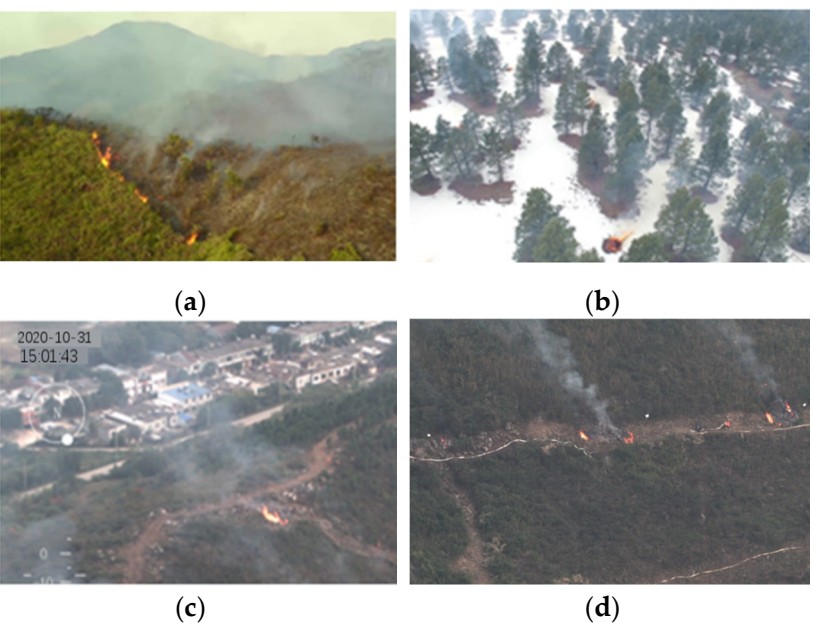

(a)　　　　　　　　　　　　　　　　　　　　　　(b)

(c)　　　　　　　　　　　　　　　　　　　　　　(d)

**Figure 2.** Pictures containing small-target fires: (**a**) small-target forest fires captured by forest video-monitoring equipment; (**b**) small-target forest fire photographed by UAV at low altitude; (**c,d**) small-target forest fires photographed by UAV at high altitude.

The details of the multi-scale forest fire data set and small-target forest fire data set are shown in Table 1.

**Table 1.** Details of the data sets.

| Dataset | Train (Multi-Scale) | Test (Multi-Scale) | Test (Small-Target Forest Fires) |
|---|---|---|---|
| Number of forest fire data set pictures | 2537 | 298 | 332 |

### 2.2. Swin Transformer

At present, there are two major challenges when applying Transformer in computer vision tasks: First, the size of the target is dynamic. Unlike NLP tasks, in which the token size is basically the same, the target size in target detection is different, and it is difficult to achieve good results with a single-level model. Second, if the image resolution is relatively high, the computation overhead of Transformer based on global self-attention is intolerable. In order to solve these problems, Swin Transformer constructs Transformer in a hierarchical way, which includes sliding-window operations. The sliding-window operation comprises a non-overlapping local window and an overlapping cross-window. The attention calculation is limited in a window, so that the locality of the CNN convolution operation can be introduced, and the calculation amount can be saved as well. Swin Transformer controls the calculation area in the unit of window, which greatly reduces the network's calculation amount.

The Swin Transformer is composed of four parts: (1) multi-layer perceptron (MLP); (2) window multi-head self-attention layer (W-MSA); (3) sliding-window multi-head self-attention layer (SW-MSA), and (4) layer normalization (LN). The backbone of the Swin Transformer is shown in Figure 3.

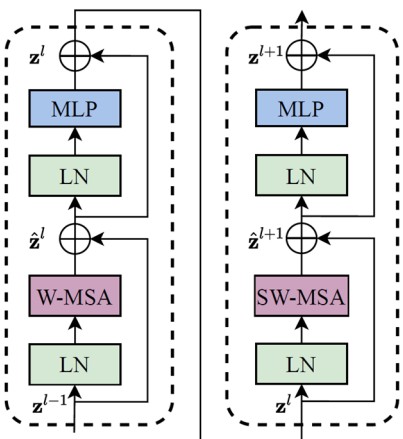

**Figure 3.** Backbone of the Swin Transformer.

It can be seen from Figure 3 that the features input to the Swin Transformer backbone network are first normalized by LN, and learned by W-MSA. Then, the residual is calculated, and the results are passed through an LN and an MLP. Finally, the residual operation is performed again to obtain the output features of this layer. SW-MSA is similar to W-MSA in structure, but the difference is that the SW-MSA requires the sliding window operation. During the whole process, the output of each step is represented in Formulas (1) to (4):

$$\hat{z}^l = W - MSA(LN(z^{l-1})) + z^{l-1} \tag{1}$$

$$z^l = MLP(LN(\hat{z}^l)) + \hat{z}^l \tag{2}$$

$$\hat{z}^{l+1} = SW - MSA(LN(z^l)) + z^l \tag{3}$$

$$z^{l+1} = MLP(LN(\hat{z}^{l+1})) + \hat{z}^{l+1} \tag{4}$$

The overall network structure of the Swin Transformer is shown in Figure 4.

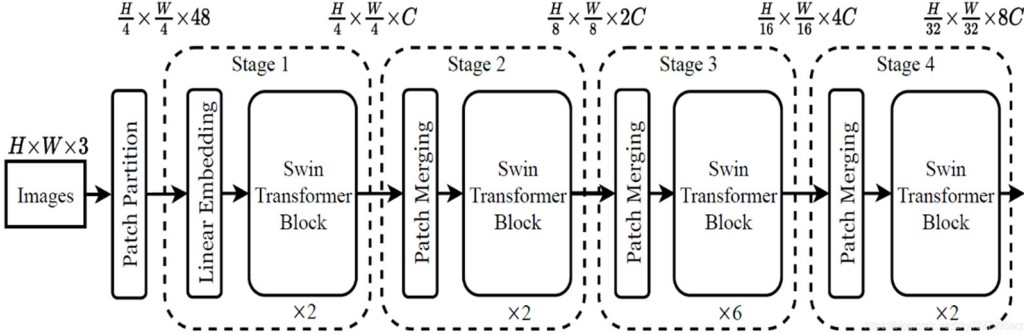

**Figure 4.** Swin Transformer overall network structure.

The self-attention mechanism is the key module of Swin Transformer, and its calculation method is shown in Formula (5). *Q*, *K*, and *V* are query, key, and value, respectively. d is the query dimension. *B* is relative position offset.

$$A = Attention\,(Q, K, V) = SoftMax\left(\frac{QK^T}{\sqrt{d}} + B\right)V \tag{5}$$

As the depth of the Swin Transformer increases, the image blocks are gradually merged to construct a hierarchical transformer, which can be used as a general visual backbone network for tasks such as image classification, target detection, and semantic segmentation, and can also be used for dense prediction tasks, improving the performance of small-target detection.

### 2.3. Mask R-CNN

Mask R-CNN is a well-known member of the R-CNN family. Its outstanding performance in target detection and semantic segmentation makes it one of the best target detection models at present.

As shown in Figure 5, a mask prediction branch is added on the basis of the Faster R-CNN, and a Feature Pyramid Network (FPN) is incorporated into the backbone (ResNet-50) for feature extraction, enabling better utilization of multi-scale information; also, the region of interest pooling layer is improved (Region of Interesting Pooling, RoI Pooling) is the ROI matching layer (Region of Interesting Align, RoI Align), and the bilinear interpolation method is used to replace the rounding method in the prediction box extraction process. The method solves the problem that the two quantification times are not matched in the application of RoI Pooling in the Faster R-CNN, improving the accuracy of the candidate frame. The Mask R-CNN image-processing method mainly comprises the following steps of: firstly, inputting an image into a residual network to extract features, generating a multi-scale feature map, carrying out side connection, and carrying out tensor addition on the feature map in each stage after double up-sampling and on the adjacent bottom layer; and then, which is sent to the Region Proposal Network (RPN) to generate candidate regions on the feature maps of different sizes, and input the candidate regions and the feature maps into RoI Align to obtain a prediction box, and finally classify and regress the prediction box to generate a high-quality instance segmentation mask of the detected object. On the whole, Mask R-CNN has been greatly improved in all aspects of performance compared with Faster R-CNN. In terms of loss function, Mask R-CNN creatively uses a multi-task loss function to minimize the value of the loss function by means of continuous learning, and finally achieves the global optimal solution.

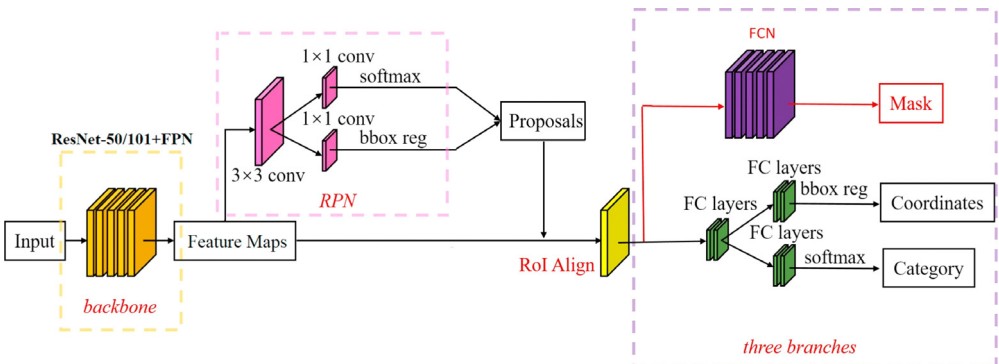

**Figure 5.** The network structure of Mask R-CNN.

### 2.4. PAFPN

Based on FPN, PAFPN adds the fusion path from bottom to top, enhancing the multi-scale fusion information of FPN. Low-level features play an important role in locating large targets, but there is a long path between high-level features and low-level features (as shown in Figure 6a), bringing great difficulty in performing accurate positioning. In order to enhance the feature pyramid with low-level accurate positioning information and shorten the information path, a bottom-up path enhancement based on FPN (shown in Figure 6b) is created by PAFPN, which can improve the feature pyramid architecture and shorten the information path by using the precise positioning signals stored in the low-level features.

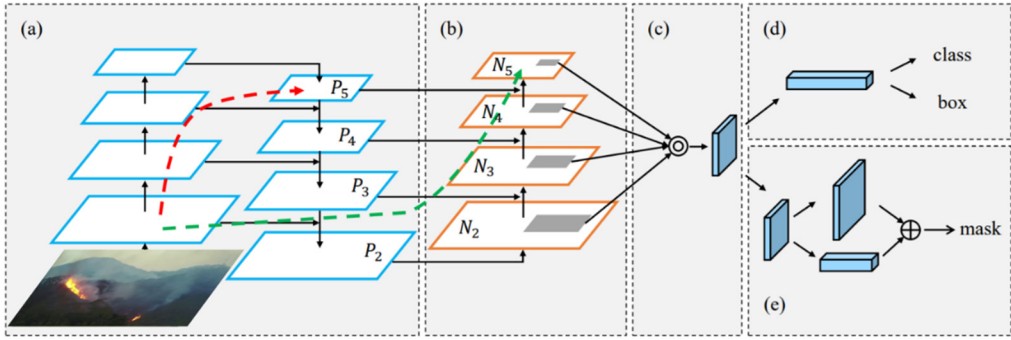

**Figure 6.** The network architecture of PAFPN. (**a**) is FPN; (**b**) is the bottom-up feature fusion layer added by PAFPN; (**c**) is the adaptive feature pooling layer; (**d**) is the bounding box prediction header of PAFPN; (**e**) is the full connection fusion layer used for prediction mask.

PAFPN creates adaptive feature pooling, as shown in Figure 6c, which is used to restore the corrupted information path between each candidate region and all feature levels, aggregating each candidate region on each feature level. The target detection and instance split-sharing network architecture of PAFPN consists of the three parts shown Figure 6a–c, so that the performance of both is improved.

### 2.5. Slicing Aided Hyper Inference

Slicing Aided Hyper Inference (SAHI) is a slice-based technology for mitigating the difficulty in detecting small objects in the inference stage. Dividing the input image into overlapping slices produces a relatively larger visibility for small targets. Similar to the idea of sliding windows, SAHI segments an image into multiple overlapping slices. In this way, the proportion of small targets in the image is relatively high.

As shown in Figure 7, the SAHI uses the slicing method in the inference process. First, the original image $I$ to be queried is sliced into $l$ number of $M*N$ overlapping patches: $P_1^I, P_2^I, P_3^I, \cdots\cdots P_l^I$. Then, resize the size of each patch while maintaining the aspect ratio. After that, object detection forward transfer is independently applied to each overlapping patch. Optional full inference (FI) can be chosen to detect larger objects by using the original image. Finally, Non-Maximum Suppression (NMS, the idea of the algorithm is to search the local maximum of the pixel and suppress the non-maximum) is used to merge the overlapping prediction results and FI results (if used) back to the original size. During NMS, boxes with Intersection over Union (IoU; IoU is the score measuring the accuracy of detecting corresponding objects in a specific data set) ratios lower than the predefined matching threshold, $T_m$, are removed. In this way, for one object, only the best bounding box is preserved.

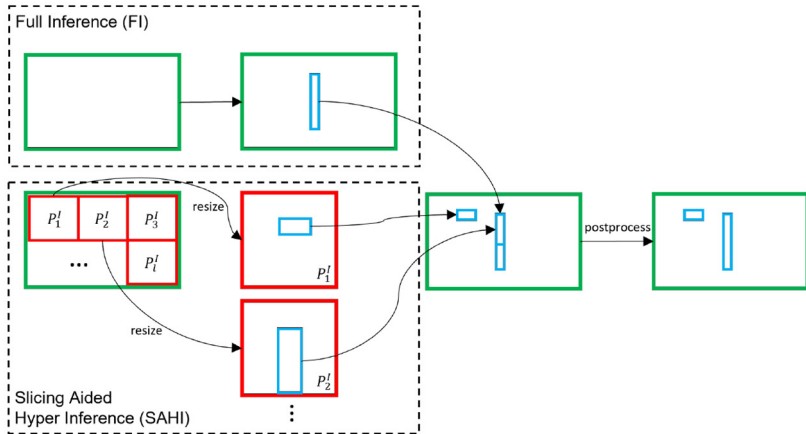

**Figure 7.** The process of slicing aided hyper inference.

### 2.6. Improved Forest Fire Detection Model STPM

We designed an improved model STPM for multi-scale forest fire target detection. Based on the original architecture of Mask R-CNN, we used Swin Transformer to replace the original backbone (ResNet-50) of Mask-RCNN, because Swin Transformer has a self-attention mechanism to (1) quickly obtain the relationship between remote elements; (2) pay attention to different areas of forest fire images; and (3) integrate the information of the whole image. Compared with ResNet-50, the improved model can capture global information to obtain a larger perception domain and context information, and therefore enhance the ability to obtain local information. Swin Transformer also performs better in dense prediction tasks and accurately detects small targets. Therefore, Swin Transformer is a promising option for small-target forest fire detection task.

However, the original FPN in Mask R-CNN may cause the loss of the main feature map information and multi-scale information. The propagation path of FPN is from top to bottom. It needs to go through dozens or even hundreds of network layers from the main feature map to the highest feature map. The amount of calculation is relatively large; thus, the difficulty in obtaining the initial image increased significantly. Therefore, we used PAFPN to solve these problems. PAFPN is improved on the basis of FPN by adding a down-sampling module and additional $3 \times 3$ convolution. By constructing a bottom-to-up feature-fusion network, the propagation path of the main feature layer is reduced, and the multi-scale fusion information of FPN is enhanced. The number of layers is less than 10; thus, the amount of calculation is small. In this way, the shallow feature information is better retained and the positioning ability of the whole feature level is improved, because the high response to the edge or the instance part is a powerful indicator for accurately positioning the instance. The structure of our improved forest fire detection model, STPM, is shown in Figure 8.

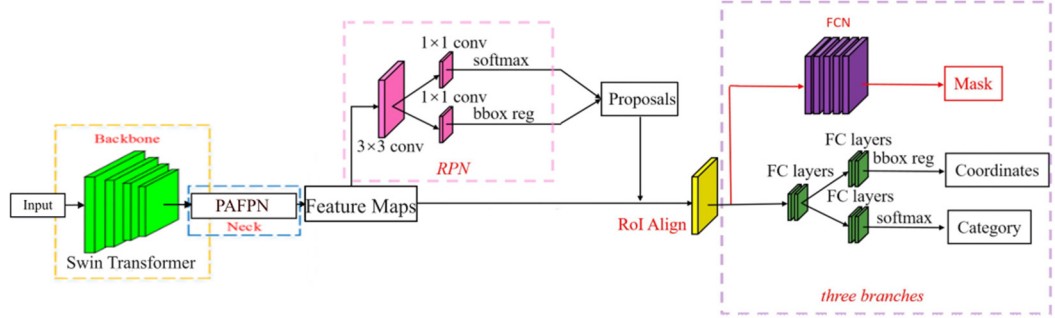

**Figure 8.** The structure of our improved forest fire detection model, STPM.

### 2.7. Small-Target Forest Fire Detection Model, STPM_SAHI

It is a challenge to detect small-target forest fires and long-distance forest fire targets based on image detection, because these small-target forest fires captured by cameras are represented by a small number of pixels and do not have sufficient detail features. So, it is difficult for traditional detectors to detect small-target forest fires. Even the accuracy of our improved high-precision forest fire detection model, STPM, is reduced. Integrating Slicing Aided Hyper Inference technology on our model solved this problem, requiring no additional training and occupying fewer computing resources. This technology provides a pipeline of slice-aided reasoning for small-target forest fires detection. The detection accuracy of small-target forest fires is greatly improved.

As shown in Figure 9, after integrating the Slicing Aided Hyper Inference framework (SAHI), SAHI divides the forest fire image to be detected, $I$, into overlapping patches: $P_1^I, P_2^I, P_3^I, \cdots \cdots P_l^I$. Then, SAHI adjusts the size of each patch to between 500 and 1000 pixels while maintaining the aspect ratio of each patch. In this process, the small-target forest fire contained in the patch will be amplified to facilitate detection. Each overlapped patch is independently processed with our improved model STPM. FI is to use the STPM model to

directly detect forest fire images that are not sliced, and it is used to detect large forest fire targets. Then the NMS algorithm is used to merge the overlapping prediction results in the slice reasoning and FI results, and the final forest fire detection results are obtained after processing. During the NMS process, the matches have boxes higher than the predefined match-prediction threshold, and for each match, detections with a lower probability of detection than what was predefined are also removed. Without SAHI technology, the detection result is only equivalent to the detection result of FI, and small-target forest fires may not be detected.

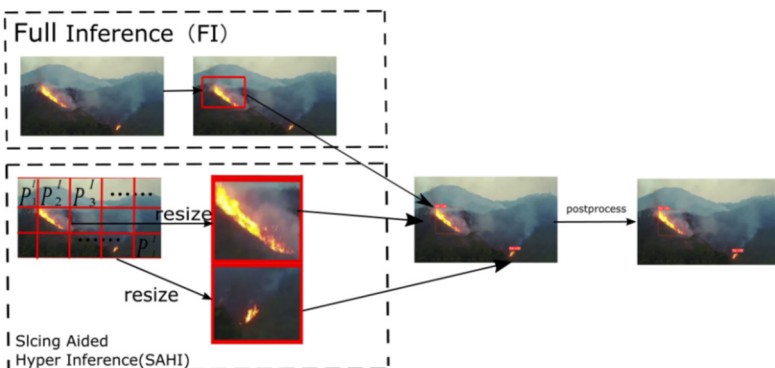

**Figure 9.** The reasoning process diagram of the Slicing Aided Hyper Inference framework for forest fire pictures.

After integrating SAHI, the overall structure and the training process of our final model STPM_SAHI for small-target forest fire detection is shown in Figure 10. First, we manually annotated the multi-scale forest fire pictures we collected and converted them into a COCO dataset format to train the improved forest fire mode STPM. Aiming at the problem that small-target forest fires are difficult to detect, SAHI technology is integrated into STPM to form the STPM_SAHI model. In total, 332 small-target forest fire pictures were used to test the detection accuracy of the model for small-target forest fires.

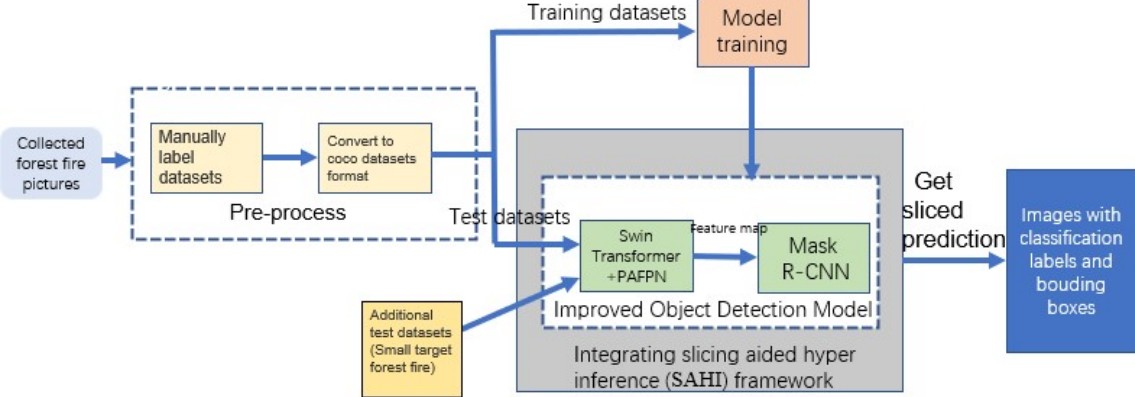

**Figure 10.** Overall structure of our small-target forest fire detection model STPM_SAHI.

### 3. Results

*3.1. Training*

The experimental environment of this paper is shown in Table 2. The training parameters of the model are shown in Table 3.

**Table 2.** Experimental conditions.

| Experimental Environment | Details |
|---|---|
| Programming language | Python 3.8 |
| Operating system | Windows 10 |
| Deep-learning framework | Pytorch 1.8.2 |
| GPU | NVIDIA RTX 3050ti |
| Deep-learning framework | Pytorch 1.8.2 |
| GPU acceleration tool | CUDA:11.1 |

**Table 3.** Training parameters of our improved model.

| Training Parameters | Details |
|---|---|
| Epochs | 300 |
| Batch-size | 16 |
| Img-size | 512 |
| Initial learning rate | 0.00125 |
| Optimization algorithm | SGD |

*3.2. Model Evaluation*

In order to verify the accuracy of the STPM model for multi-scale forest fire targets and the accuracy of the STPM_SAHI model for small-target forest fires, Microsoft COCO evaluation metrics, recognized as the most authoritative in the field of target recognition, were used in this paper. As shown in Table 4, the metrics include the detection accuracy of targets with different area sizes.

**Table 4.** Microsoft COCO standard—commonly used in object recognition tasks to evaluate the precision and recall of a model at multiple scales. The unit of AP and AR is percentage.

| Average Precision (AP) | |
|---|---|
| $AP_{0.5}$ | AP at IoU = 0.5 |
| AP Across Scales: | |
| $AP_S$ | AP for small target (Size < $32^2$) |
| $AP_M$ | AP for medium target ($32^2$ < Size < $96^2$) |
| $AP_L$ | AP for large target (Size > $96^2$) |
| Average Recall (AR) | |
| $AR_{0.5}$ | AR at IoU = 0.5 |
| AR Across Scales: | |
| $AR_S$ | AR for small target (Size < $32^2$) |
| $AR_M$ | AR for medium target ($32^2$ < Size < $96^2$) |
| $AR_L$ | AR for large target (Size > $96^2$) |

Precision (*P*) measures the proportion of all positive predictions that are correct. In the forest fire detection task, recall represents the number of correctly predicted forest fire images (*TP*) over the total number of images predicted to be forest fire (*TP* + *FP*), as shown in Formula (6).

$$P = \frac{TP}{TP + FP} \tag{6}$$

Recall (*R*) measures the proportion of all predicted samples where the prediction is positive. In forest fire detection task, the precision represents the number of forest fire images correctly predicted by the model (*TP*) and accounts for all predicted images (*TP* + *FP*), as shown in Formula (7).

$$R = \frac{TP}{TP + FN} \tag{7}$$

The Average Precision (*AP*) in Table 4 is the area enclosed by the *P–R* curve. *P* is the accuracy and *R* is the recall. The IoU threshold is typically 0.5. Generally, the larger the value is, the better the model is, and vice versa. Like *AP*, the average recall (*AR*) is also a numerical indicator that can be used to evaluate the performance of forest fire detection models. *AR* can be calculated as twice the area under the recall–IoU curve. The calculation formulas of *AP* and *AR* are shown in Formulas (8) and (9).

$$AP = \sum_{i=1}^{n-1} (R_{i+1} - R_i)P(R_{i+1}) \tag{8}$$

$$AR = 2\int_{0.5}^{1} R(o)do \tag{9}$$

The variable $o$ in the *AR* calculation formula represents the IoU between the prediction box and the ground truth box.

The process of the experiment is described as follows: First, the collected forest fire data set is used to train our model STPM to learn the characteristics of the forest fire targets at different scales. Then, evaluate the detection accuracy of our model STPM for forest fire at different scales on the test set and train it on the mainstream target detection model with the same data set, and compare their recognition accuracy with our model for forest fire targets at different scales. We also performed ablation experiments. The experimental results are shown in Table 5.

**Table 5.** Experimental results—detection accuracy of forest fires with different scales, using the Microsoft COCO evaluation indicators we introduced above.

| Model | $AP_{0.5}$ (%) | $AP_S$ (%) | $AP_M$ (%) | $AP_L$ (%) | $AR_{0.5}$ (%) | $AR_S$ (%) | $AR_M$ (%) | $AR_L$ (%) |
|---|---|---|---|---|---|---|---|---|
| YOLOv5 | 82.5 | 36.0 | 48.7 | 66.0 | 69.2 | 48.0 | 59.0 | 76.0 |
| EfficientDet | 84.5 | 36.3 | 50.2 | 64.1 | 68.6 | 49.8 | 64.2 | 73.0 |
| ResNet-50 + Mask R-CNN | 84.5 | 30.3 | 45.9 | 64.2 | 66.1 | 41.6 | 59.3 | 72.3 |
| ResNet-50 + PAFPN + Mask R-CNN | 85.7 | 34.2 | 47.0 | 65.6 | 67.1 | 44.1 | 60.6 | 73.0 |
| Swin Transformer + Mask R-CNN | 87.3 | 40.2 | 48.0 | 67.1 | 69.2 | 52.7 | 60.9 | 74.7 |
| Swin Transformer + PAFPN + Mask R-CNN(STPM, ours) | 89.4 | 42.4 | 53.7 | 67.9 | 71.2 | 56.1 | 67.0 | 75.0 |

Then 332 additional ultra-small-target forest fire images were tested on our models to compare the detection accuracy of ultra-small forest fires before and after the integration of the SAHI framework. The results are shown in Table 6.

**Table 6.** Experimental results—detection accuracy of small-target forest fire after integrating SAHI.

| Model | Small-Target Forest Fires $AP_{0.5}$ (%) |
|---|---|
| YOLOv5 | 22.7 |
| EfficientDet | 27.4 |
| ResNet-50 + Mask R-CNN | 33.9 |
| ResNet-50 + PAFPN + Mask R-CNN | 36.7 |
| Swin Transformer + Mask R-CNN | 46.2 |
| Swin Transformer + PAFPN + Mask R-CNN(STPM, ours) | 50.9 |
| STPM + SAHI(STPM_SAHI, ours) | 59.0 |

We also tested the detection speed of the STPM_SAHI on edge-side devices. STPM_SAHI detected forest fire videos of different resolutions in real time. The time consumption of each frame during image detection is shown in Table 7.

**Table 7.** Time consumption of STPM_SAHI for real-time detection of each frame of forest fire videos with different resolutions (video resolution refers to the number of pixels contained in a video in a certain area).

| Video Resolution (Pixel) | Time |
|---|---|
| 426 × 240 | 52.01 ms |
| 640 × 360 | 181.04 ms |
| 720 × 416 | 196.04 ms |
| 845 × 480 | 208.01 ms |

### 3.3. Detection Performance and Analysis

After the experiment, we can see that although YOLOv5 and EfficientDet are widely used in target detection, their accuracy toward forest fire target detection is not as high as our model, STPM, and STPM is better than YOLOv5 and EfficientDet in detecting small-sized, medium-sized, and large-sized forest fire targets. Especially for small fires with pixels less than $32^2$, our model is significantly better than YOLOv5 and Efficient Det in average precision ($AP_S$). Therefore, our improved model STPM has a higher detection accuracy for small-target forest fires. After integrating the SAHI framework into STPM, our model STPM_SAHI further improves the forest fire detection accuracy ($AP_{0.5}$) of ultra-small targets by 8.1, 36.3 higher than YOLOv5 and 31.6 higher than EfficentDet. Based on the experiment results, it can be concluded that our STPM_SAHI model is suitable for the detection of small-target forest fires and the detection accuracy toward forest fire targets at different scales is also very high. Some of the test results are shown in Figures 11–14.

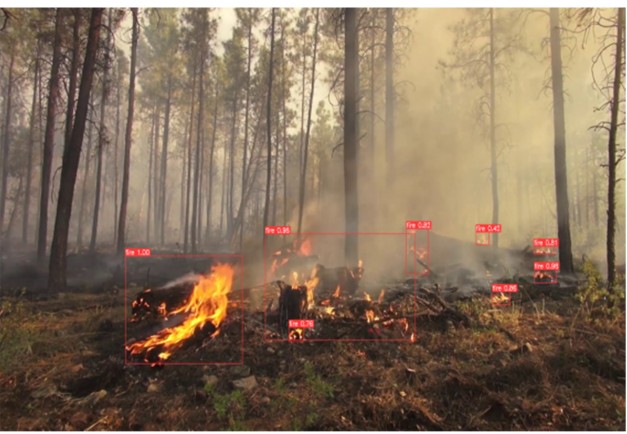

**Figure 11.** Detection result of small-target forest fires and large-target forest fires using STPM_SAHI.

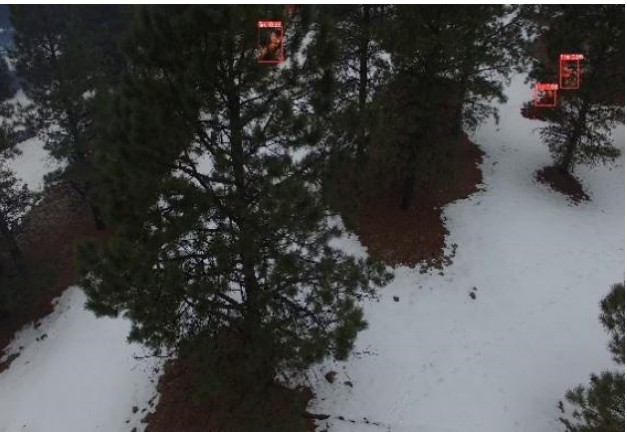

**Figure 12.** Detection result using STPM_SAHI on a snowy day.

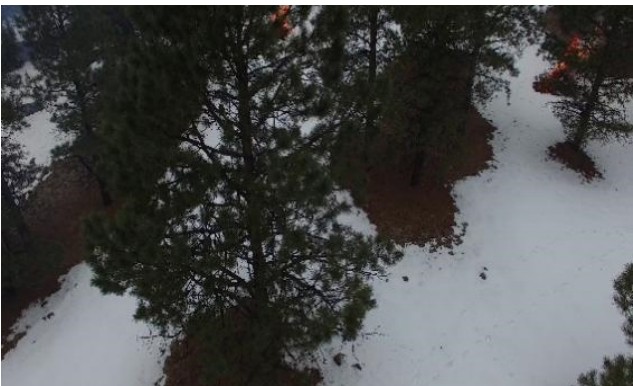

**Figure 13.** Result detected with YOLO.

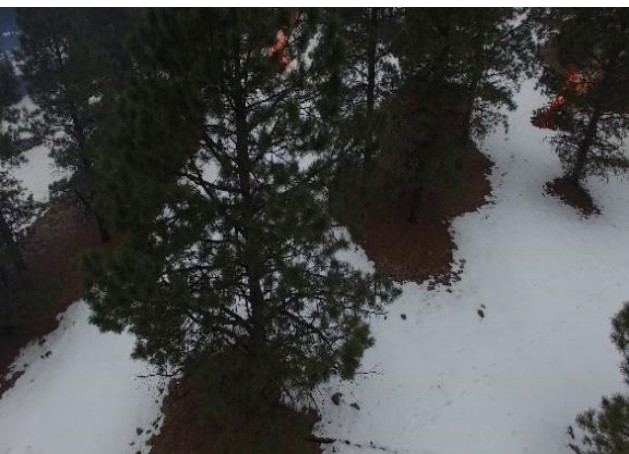

**Figure 14.** Result detected with EfficientDet.

As shown in Figure 11, we used STPM_SAHI to detect an image containing large target forest fires and small-target forest fires without any missed detection. In Figure 12, all fires, which are ultra-small in size, can be detected by our model STPM_SAHI, but Yolov5 and EfficientDet failed to detect them (as shown in Figures 13 and 14).

Figure 15 shows the detection of forest fire targets in very small pixels in the non-integrated Slicing Aided Hyper Inference framework, and the model missed the fire regions. However, as shown in Figure 16, after integrating the SAHI framework, all fires were detected by STPM_SAHI, with a confidence level of more than 0.9. It is also worth mentioning that the firefighters photographed from high altitude are very similar to forest fires, but they were not mis-detected by STPM_SAHI.

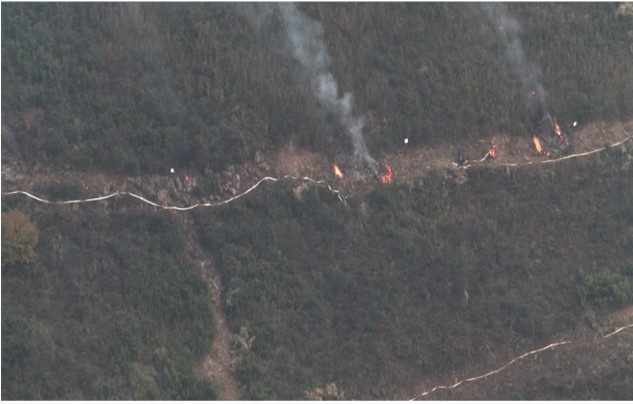

**Figure 15.** Detection result using STPM.

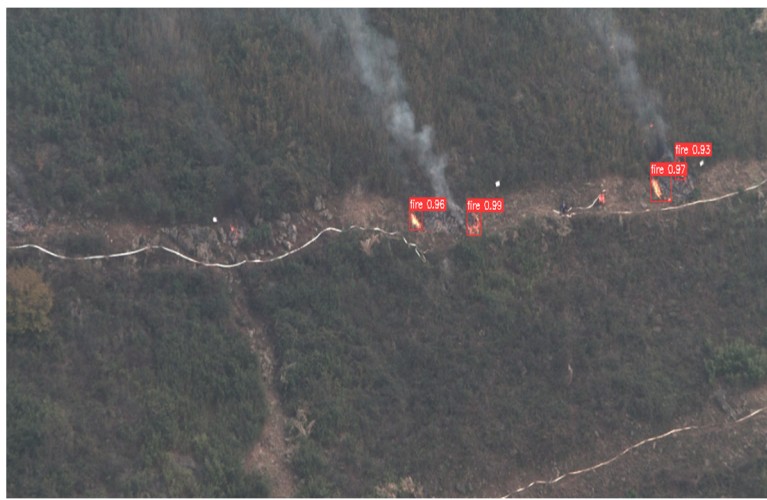

**Figure 16.** Detection result using STPM_SAHI.

STPM_SAHI also meets the real-time detection requirements. Experiments show detection delays only in milliseconds, and the detection delay will not affect the timely detection of forest fires. The real-time video recognition result is shown in Figure 17.

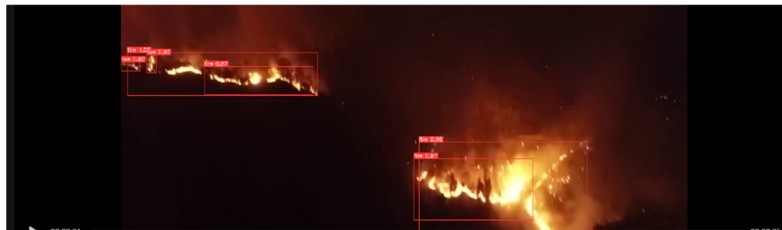

**Figure 17.** Real-time detection result using a forest fire video.

## 4. Discussion and Conclusions

It is very important to detect forest fires, especially to detect small-target forest fires in time and prevent their spread. In recent years, most forest fire detection models have been based on convolutional neural networks (CNNs). Compared with our forest fire model that use Swin Transformer to extract features, these models directly obtain global context information, is full of challenges, and do not have strong global modeling capabilities—the limitations of the convolution operation. Another problem is that small-target forest fires are very small and do not have sufficient, detailed features, which are difficult to be detected by traditional target detection models. Even improved, high-precision models found it difficult to detect forest fire targets with very small pixels. We introduced Slicing Aided Hyper Inference technology for the first time in the field of forest fire detection to solve this problem. This technology provides a pipeline of slice-aided reasoning for small-target forest fire detection, which greatly improves the detection accuracy of small-target forest fires.

In this paper, a small-target forest fire detection mode is designed. Firstly, a multi-scale forest fire detection model STPM was designed. The self-attention mechanism of the backbone network of Swin Transformer was used to obtain a greater perception domain and context information, capture the global information of forest fires, and enhance the extraction ability of local forest fire features, making the model pay more attention to and fully learn the characteristics of each forest fire.

Secondly, the Mask R-CNN detection framework is improved, PAFPN is used as the feature-extraction network, and a down-sampling module and an additional convolution module were added to reduce the propagation path of the main feature layer and eliminate the influence of down-sampling fusion, so as to enhance the positioning ability of the whole feature layer. After training, the detection accuracy toward different-scale forest fire

targets is satisfactory. The average precision ($AP_{0.5}$) reaches 89.4, and the detection accuracy toward different-scale forest fire targets is better than the mainstream target detection models, especially for small-target forest fires with pixels less than $32^2$.

Then, the Slicing Aided Hyper Inference technology was integrated with our improved forest fire detection model, STPM, to form the STPM_SAHI model. The accuracy of the model was verified by 332 small-target forest fire images. The detection accuracy of the small-target forest fires is significantly improved when using STPM_SAHI. The average precision ($AP_{0.5}$) in detecting small-target forest fires in our test set increased by 8.1, reaching 59.0. Testing on the same small-target forest fire data set, the average precision ($AP_{0.5}$) of YOLOv5 and EfficientDet was only 22.7 and 27.4, respectively. Through ablation experiments, we have proved the effectiveness of each module of the improved forest fire detection model, and the forest fire detection accuracy was significantly better than that of the mainstream models. Our model can also detect forest fire targets with very small pixels that mainstream target detection models find difficult to detect. Our model is highly suitable for small-target forest fire detection. The detection accuracy of forest fire targets at different scales is also very high and meets the needs of real-time forest fire detection.

STPM_SAHI can be deployed at the edge of UAVs equipped with cameras, watchtowers monitored by video, and various video-monitoring platforms, for real-time detection of forest fires. Or, it can be deployed in the cloud to process footage transmitted by video surveillance equipment in real time and automatically detect forest fires. However, like other models, the processing speed of the model for ultra-high-resolution video may decline. In the next stage, we will solve this problem and further improve the accuracy of the model.

**Author Contributions:** J.L. devised the programs and drafted the initial manuscript; H.L. and F.W. designed the project and revised the manuscript. All authors have read and agreed to the published version of the manuscript.

**Funding:** This work was supported by the Key Research and Development plan of Jiangsu Province (Grant No. BE2021716), the Jiangsu Modern Agricultural Machinery Equipment and Technology Demonstration and Promotion Project (NJ2021-19), the Natural Science Foundation of Jiangsu Province (BK20191393), the National Natural Science Foundation of China (32101535), and the Jiangsu Postdoctoral Research Foundation (2021K112B).

**Data Availability Statement:** This work uses the publicly available dataset FLAME, see reference [34] for data availability.

**Conflicts of Interest:** The authors declare no conflict of interest.

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
