# Peer review of "STPM_SAHI: A Small-Target Forest Fire Detection Model Based on Swin Transformer and Slicing Aided Hyper Inference"

_forests, doi:10.3390/f13101603_

Round 1
Reviewer 1 Report (Previous Reviewer 1)
The author adds some experiments and explanations to make the sample manuscript richer, more logically coherent, and more convincing, proving that the method does have excellent performance and certain innovation, but there are still some grammatical errors, after modification can be received.
Author Response
Authors’ Reply to Reviewers
Thanks, the reviewer for giving us the helpful comments and suggestions, which helps us to improve the quality of the manuscript. In the revised version, we have revised their concerns and changed made in the revised manuscript are red in color. The followings are our replies to the reviewers’ comments: the words in blue are the questions of the reviewers, and the words in black are our replies.
Reviewer #1:
The author adds some experiments and explanations to make the sample manuscript richer, more logically coherent, and more convincing, proving that the method does have excellent performance and certain innovation, but there are still some grammatical errors, after modification can be received.
Reply to Review 1: This is a good question and we have made necessary changes. We carefully checked the grammar and spelling errors again to make the quality of the article higher.
Changes Made: According to this comment, we have made necessary changes, which are colored red in the revised manuscript.

Reviewer 2 Report (New Reviewer)
The title sound good, does not need any revisions. The title and focus of the manuscript should be of interest of a wide group of experts dealing with forest fire prevention, detection and preparedness.
The abstract is well-written, it includes all the aspects needed. However, there are minor revisions, regarding text formatting, required.
The introduction section clearly presents state-of-the-art in the forest fore detection. The publications mentioned are current and there is no need for further supplementing this section. However, I would like to recommend to move the text in lines 100-127 to methodology or to the conclusions section.
Material and methodology description well-chosen, clearly described and repeatable.
In both sections, there is the revision of text formatting required. English proof-reading is also recommended.
Results are clearly described, but the separation of tables should be enhanced. However, the units in which the accuracy or precision, image resolution is expressed is needed to be introduced in each table and also in the text.
In the Conclusion and Discussion section, the discussion is missing. In the Discussion, the comparison of own results with the findings of other authors (must be introduced).
Author Response
Authors’ Reply to Reviewers
Thanks, the reviewer for giving us the helpful comments and suggestions, which helps us to improve the quality of the manuscript. In the revised version, we have revised their concerns and changed made in the revised manuscript are red in color. The followings are our replies to the reviewers’ comments: the words in blue are the questions of the reviewers, and the words in black are our replies.
Reviewer #2:
- The title sound good, does not need any revisions. The title and focus of the manuscript should be of interest of a wide group of experts dealing with forest fire prevention, detection and preparedness.
Reply to Question 1: Thank you for your affirmation of our title. We followed your advice and did not modify the title.
- The abstract is well-written, it includes all the aspects needed. However, there are minor revisions, regarding text formatting, required.
Reply to Question 2: This is a good question and we have made necessary changes. We have made some minor modifications to the text problem.
Changes Made: According to this comment, we have made necessary changes, which are colored red in the revised manuscript.
- The introduction section clearly presents state-of-the-art in the forest fore detection. The publications mentioned are current and there is no need for further supplementing this section. However, I would like to recommend to move the text in lines 100-127 to methodology or to the conclusions section.
- Reply to Question 3: This is a good question and we have made necessary changes. Following your suggestion, we revised the text in lines 100-127 and moved the relevant content to methodology section.
Changes Made: According to this comment, we have made necessary changes, which are colored red in the revised manuscript. As shown below and marked in red.
Another problem is that small target forest fires are very small and lack sufficient details, so they are difficult to be detected.To solve this problem, we integrate our improved forest fire target detection model with Slicing Aided Hyper Inference (SAHI) [30] technology which provides a pipeline of slice aided reasoning for small target forest fires detection and greatly improves the detection accuracy of small target forest fires.
2.Materials and Methods
The contributions of this paper are list as follows:
1)We use Swin Transformer replaces the original backbone (ResNet-50) of Mask R-CNN to makes full use of its self-attention mechanism, obtain larger receptive field and context information by capturing global information and strengthen the ability of local information acquisition. The model pays more attention to and fully learns the characteristics of forest fires, which improves the detection performance of small target forest fire images captured by cameras, and has better detection performances on intensive forest fires.
2)We use PAFPN [31] to replace the Mask R-CNN detection framework’s original feature fusion network FPN [32]. Based on FPN, PAFPN adds a down-sampling module and an additional 3 × 3 convolution to build a bottom-up feature fusion network to reduce the propagation path of the main feature layer and eliminate the impact of down-sampling fusion. The positioning capability of the whole feature hierarchy is enhanced.
3) After the model training, we integrated Slicing Aided Hyper Inference technology with our improved forest fire detection model, which solved the problem that the target pixels in small target forest fires are small in proportion and lack sufficient details to be detected, while maintaining low computational resource requirements.
……
The self-attention mechanism is the key module of Transformer, and its calculation method is shown in formula (5).
(5)
As the depth of the Swin Transformer increases, the image blocks are gradually merged to construct a hierarchical transformer, which can be used as a general visual backbone network for tasks such as image classification, target detection and semantic segmentation, and can also be used for dense prediction tasks, and the detection performance of small targets is improved.
- Material and methodology description well-chosen, clearly described and repeatable. In both sections, there is the revision of text formatting required. English proof-reading is also recommended.
Reply to Question 4: This is a good question and we have made necessary changes. We carefully revised the text format and checked the English grammar and spelling errors.
Changes Made: According to this comment, we have made necessary changes, which are colored red in the revised manuscript.
- Results are clearly described, but the separation of tables should be enhanced. However, the units in which the accuracy or precision, image resolution is expressed is needed to be introduced in each table and also in the text.
Reply to Question 5: This is a good question and we have made necessary changes. We have strengthened the separation of tables. We added the introduction of relevant units. And the names of relevant units have also been added in the tables. The calculation formula of some indicators is also supplemented.
Changes Made: According to this comment, we have made necessary changes, which are colored red in the revised manuscript. As shown below and marked in red.
The calculation formulas of AP and AR are shown in formula (8) and formula(9).
(8)
(9)
Table 4 Microsoft COCO Standard — Commonly used in object recognition tasks to evaluate the precision and recall of a model at multiple scales. The unit of AP and AR is percentage.
Table 5 Experimental results - Detection accuracy of forest fires with different scales. Use the Microsoft COCO evaluation indicators we introduced above.
Table 6 Experimental Results- Detection accuracy of small target forest fire after integrating SAHI.
Table 7 Time consuming of STPM_SAHI for real-time detection of forest fire video with different resolutions (Video resolution refers to the number of pixels contained in a video in a certain area).
6.In the Conclusion and Discussion section, the discussion is missing. In the Discussion, the comparison of own results with the findings of other authors (must be introduced).
Reply to Question 6: This is a good question and we have made necessary changes. In the conclusion and discussion part, we have added some discussion and added some comparisons with the models proposed by other authors in recent years.
Changes Made: According to this comment, we have made necessary changes, which are colored red in the revised manuscript. As shown below and marked in red.
- Discussion and Conclusions
It is very important to detect forest fires, especially to detect small target forest fires in time and prevent their spread. In recent years, most forest fire detection models are based on convolutional neural networks (CNNs). Compared with our forest fire model that use Swin Transformer to extract features, these models directly obtain global context information is full of challenges and do not have strong global modeling capabilities for the limitations of convolution operation. Another problem is that small target forest fires are very small and do not have sufficient detail features, which are difficult to be detected by traditional target detection models. Even the improved high-precision model is difficult to detect forest fire targets with very small pixels. We introduced Slicing Aided Hyper Inference technology for the first time in the field of forest fire detection to solve this problem. This technology provides a pipeline of slice aided reasoning for small target forest fire detection, which greatly improves the detection accuracy of small target forest fires.
In this paper, a small target forest fire detection mode is designed. Firstly, a multi-scale forest fire detection model STPM is designed. The self-attention mechanism of the backbone network of Swin Transformer is used to obtain greater perception domain and context information, capture the global information of forest fire, and enhance the extraction ability of local forest fire features, making the model pay more attention to and fully learn the characteristics of forest fire.
Secondly, the Mask R-CNN detection framework is improved, PAFPN is used as the feature extraction network, and a down sampling module and an additional convolution module are added to reduce the propagation path of the main feature layer and eliminate the influence of down sampling fusion, so as to enhance the positioning ability of the whole feature layer. After training, the detection accuracy of different scale forest fire targets is satisfactory. The average precision (AP0.5) reaches 89.4, and the detection accuracy of different scale forest fire targets is better than the mainstream target detection models. Especially the small forest fire targets with pixels less than 322.
Then, the Slicing Aided Hyper Inference technology is integrated with our improved forest fire detection model STPM to form STPM_ SAHI model. The accuracy of the model was verified by 332 small target forest fire images. The detection accuracy of the small target forest fires is significantly improved when using STPM_ SAHI. The average precision (AP0.5) of detecting small target forest fires in our test set is increased by 8.1, reaching 59.0. Test on the same small target forest fire test set, the average precision (AP0.5) of YOLOv5 and EfficientDet is only 22.7 and 27.4. Through ablation experiments, we have proved the effectiveness of each module of the improved forest fire detection model, and the detection accuracy of forest fire is significantly better than that of the mainstream models. Our model can also detect forest fire targets with very small pixels that mainstream target detection models are difficult to detect. Our model is very suitable for small target forest fire detection. The detection accuracy of forest fire targets with different scales is also very high and meets the needs of real-time forest fire detection.
STPM_ SAHI can be deployed at the edge of UAVs equipped with cameras, watchtowers monitored by video and various video monitoring platforms for real-time detection of forest fires. Or it can be deployed in the cloud to process footage transmitted by video surveillance equipment in real time and automatically detect forest fires. However, like other models, the processing speed of the model for ultra-high-resolution video may decline. In the next stage, we will solve this problem and further improve the accuracy of the model.

This manuscript is a resubmission of an earlier submission. The following is a list of the peer review reports and author responses from that submission.
Round 1
Reviewer 1 Report
This paper proposed a model for detecting forest fire targets with very small pixels from the perspective of high-altitude UAV. For forest fire targets of different scales, experiments show that the accuracy of the method is higher than the current mainstream network. However, the novelty and scientific impact of this work appears to be limited. The authors only verified the accuracy of the method, but did not take into account the time, feature extraction effect, etc.. We could not fully know the advantages of this approach. Besides, there are some other issues.
1.In lines 10 to 14, the authors claimed that “We use Swin Transformer backbone network as the feature extraction network, and its self-attention mechanism is used to capture the global information of forest fire, which strengthens the ability of local forest fire feature extraction, and improves the ability of feature extraction of small target forest fire from the perspective of UAV”, there are the following two issues.
First, the use of the self-attention mechanism to capture global information was mentioned in the previous article, but the latter said that the ability to extract local information was strengthened. The logic was chaotic and the meaning was unclear.
Second, the authors did not use experiments to prove that Swin Transformer has better extraction for small targets.
2. Forest fire detection has certain requirements on time, and it is suggested that the authors use experiments to prove whether the method can meet the real-time performance.
3. The authors only compared with the current mainstream object detection algorithms, but did not use ablation experiments to demonstrate the importance of each module.
4. It is recommended that the author further introduce the research progress of small target detection in the introduction.